

# The prognostic value of preoperative neoindices consisting of lymphocytes, neutrophils and albumin (LANR) in operable breast cancer: a retrospective study

Yuan Wang[1,*], Jiaru Zhuang[1,*], Shan Wang[1], Yibo Wu[1] and Ling Chen[2]

[1] Human Reproductive and Genetic Center, Affiliated Hospital of Jiangnan University, Wuxi, China
[2] Department of Breast Surgery, Affiliated Hospital of Jiangnan University, Wuxi, China
[*] These authors contributed equally to this work.

## ABSTRACT

**Background**. Preoperative inflammatory factors and nutritional status are strongly associated with the prognosis of a variety of cancers. We explored the relationship between preoperative lymphocytes, neutrophils and albumin (LANR) and progression-free survival in breast cancer patients.

**Methods**. The clinical and follow-up data of 200 breast cancer patients were retrospectively analyzed in this study, and the value of LANR was determined as follows: LANR, lymphocytes × albumin/neutrophils. ROC curves, COX proportional risk regression analysis and subgroup analysis were used to assess the prognostic value of LANR in progression-free survival of breast cancer patients.

**Results**. The median age of the patients was 55.5 years (range 50–62 years). The median follow-up time was 46 months (range 33–55 months). In progression-free survival, the area under the LANR curve was 0.748 and the HR (95% CI) was 0.035 (0.679–0.817). LANR was associated with age ($p = 0.02$), positive axillary lymph nodes ($p < 0.001$), TNM stage ($p < 0.001$) and human epidermal growth factor receptor 2($p = 0.004$). The results indicated that preoperative LANR may be a reliable predictor of progression-free survival in patients with operable breast cancer.

**Conclusion**. LANR may be an essential predictor for breast cancer patients and provides a therapeutic basis for clinicians and patients.

# INTRODUCTION

Breast cancer is an issue of public health significance and is the most commonly diagnosed malignancy in women throughout the world (*Ferlay et al., 2015*). Epidemiological data reveal that it ranks first in the Chinese female population in terms of incidence (*Sung et al., 2021*). While breast cancer has a relatively satisfactory prognosis compared to other tumors, such as gastrointestinal tumors and lung cancer, survival outcomes for

Corresponding authors
Yibo Wu, 9862016107@jiangnan.edu.cn
Ling Chen, rainbow_lyn@163.com

patients with advanced or distant metastatic breast cancer remain very poor (*Shankaran et al., 2001*). Currently, traditional prognostic considerations include the patient's age, TNM (tumor-lymph node-metastasis) stage, histological grading, estrogen receptor (ER), progesterone receptor (PR), human epidermal growth factor receptor 2 (c-erbB2 or HER2) status have been used as prognostic elements in breast cancer, but reliable and economical prognostic models are still lacking (*D'Eredita' et al., 2001*).

It is well established that inflammatory status and nutritional factors are crucial for cancer prognosis (*McGovern et al., 2022*). To predict the prognosis of breast cancer, peripheral blood biomarkers representing inflammation and tumor load have been investigated several times (*Tokunaga et al., 2015*). It is currently widely accepted that systemic inflammatory factors, such as neutrophil-to-lymphocyte ratio (NLR), monocyte-to-lymphocyte ratio (MLR) (*Hua et al., 2020*), platelet-to-lymphocyte ratio (PLR), and blood biochemical indicators related to nutritional status, such as C-reactive protein (CRP) and albumin (ALB) levels, are used to reflect the nutritional and inflammatory status of patients (*Liang et al., 2021*). Additionally, there is controversy regarding the optimal threshold values for these inflammatory biomarkers and nutritional status to predict prognosis. Since blood biochemical indices are economical and readily available (*Antonio et al., 2015*), while inflammatory markers and nutritional status may affect the prognosis of patients with operable breast cancer; therefore, our study was to investigate the prognostic significance of LANR index in operable breast cancer. Previous studies have shown that LANR is associated with prognosis in resectable colorectal, nasopharyngeal, and cervical cancers (*Zhang et al., 2024*; *Wang et al., 2023*; *Liang et al., 2021*). But to date, no study has shown the prognostic value of LANR in operable breast cancer.

Accordingly, this study retrospectively analyzed the preoperative blood biochemical indices of 200 breast cancer patients and comprehensively assessed the prognostic value of survival in LANR using subject operating characteristic curves (ROC) and Kaplan–Meier survival curves.

## MATERIALS & METHODS

### Research design

We retrospectively collected clinical and pathologic information on 200 female breast cancer patients who underwent their first surgical treatment for breast cancer from January 2015 to December 2019 at the Affiliated Hospital of Jiangnan University. The cancer stage of each patient was determined according to the eighth edition of the American Joint Committee on Cancer (AJCC) staging manual. The inclusion criteria were as follows: (1) all patients had a pathologic diagnosis of primary breast cancer; (2) patients with histologically confirmed non-metastatic invasive female breast cancer; (3) all underwent radical surgical resection for the first time; (4) had complete clinicopathologic and laboratory data. Exclusion criteria: (1) patients with a history of malignancy or a combination of other primary tumors; (2) patients with acute and chronic inflammation and infection prior to surgery; (3) patients who had received neoadjuvant therapy such as radiotherapy prior to surgery; This study was approved by the Medical Ethics Committee of the Affiliated

Hospital of Jiangnan University (JNMS04202301072). The Ethics Committee waived the requirement for informed consent because all data were anonymously summarized. The research was conducted in accordance with the principles of the Declaration of Helsinki.

## Data collection

We collected preoperative blood biochemical indices and pathological information of breast cancer patients through hospital electronic medical records and clinical data. All patients' blood was collected the week before the surgery. Body mass index (BMI) was classified as <18.5, 18.5–23.9, and >23.9. Cancer stage (including tumor size, positive axillary lymph nodes, and TNM) was assessed for each patient according to the American Joint Committee on Cancer (AJCC) staging manual, eighth edition (*Amin et al., 2017*). Serum biochemical tests, including laboratory data (lymphocytes, neutrophils and albumin), were performed at the preoperative baseline follow-up. Follow-up information was obtained from outpatient review and telephone follow-up. The endpoint of this study was progression-free survival, defined as the time from postoperative pathological diagnosis to medical imaging showing distant metastases (*Martini et al., 2022*).

## Statistical analysis

SPSS 26.0 software was used for statistical analysis. Independent samples $t$-test or Wilcoxon test was used to compare differences between groups of continuous variables. The chi-square test or Fisher's exact test was used to calculate differences between categorical variables. With disease progression or death as the endpoint, the Youden index was calculated using the subject operating characteristic (ROC) curve and the corresponding area under the ROC curve (AUC) values were calculated. In survival analysis, Kaplan–Meier survival curves and log-rank tests were used to compare survival differences between groups classified by dichotomous biochemical indicators. Cox proportional risk regression models were used for univariate and multivariate regression analyses. Subgroup analyses were performed to show the prognostic correlation between patients with different characteristics and the new index. All analyses were two-sided, and $p$ values <0.05 were considered statistically significant.

## RESULTS

### Patient characteristics

A total of 200 breast cancer patients, all of whom eventually underwent surgery, were included in this study, and the demographic and clinicopathological characteristics are shown in Table 1. The median age of all patients was 55.5 years (range, 50-62 years). According to the AJCC classification, there were 82 patients (41%) with stage I, 70 patients (35%) with stage II, and 48 patients (24%) with stage III. Seven (3.5%) were underweight, 94 (47%) were normal weight, and 99 (49.5%) were overweight. In terms of tumor size, 101 patients (50.5%) had T1, 96 patients (48%) had T2, and three patients (1.5%) had T3. A total of 126 patients (63%) had negative axillary lymph nodes, 38 patients (19%) had one to three positives, 21 patients (10.5%) had four to nine positives, and 15 patients (7.5%) had more than 10 positives. A total of 130 patients (65%) had positive ER status.

The ER status was positive in 65% of patients and negative in 70 (35%). 98 patients (94%) had positive PR status and 102 (51%) were negative. For Her-2 status, 163 (81.5%) showed positive and 37 (18.5%) negative. A total of 69 patients (34.5%) relapsed and 131 patients (65.5%) did not relapse. The median follow-up time was 46 months.

### Prognostic value of LANR in progression-free survival

The area under the ROC curve for PFS and the inflection points for lymphocytes neutrophils, albumin, and LANR are listed in Table 2. Based on the ROC curve, we found the best area under the LANR curve at 0.748 (Fig. 1). We divided LANR into high level ($n = 87$, 43.5%) and low level ($n = 113$, 56.5%) groups based on critical values (Table 3). Kaplan–Meier survival curves showed that patients with high levels of LANR had longer progression-free survival (Fig. 2). Univariate analysis showed that lymphocytes, neutrophils, albumin, LANR, age, TNM stage, tumor size and positive axillary lymph nodes were significantly associated with PFS (progression-free survival) in breast cancer patients (all $p < 0.05$; Table 2). Multivariate analysis showed that albumin, LANR, age, TNM stage and positive axillary lymph nodes were associated with PFS. The results suggest that LANR may be a reliable predictor of progression-free survival in breast cancer patients.

## DISCUSSION

In this study, our research confirms the use of a new indicator: LANR, which is a scoring system based on systemic inflammatory response and nutritional status, and the results show that this indicator is associated with the prognosis of breast cancer patients. To our knowledge, this is the first study to investigate the prognostic value of LANR in patients with breast cancer.

LANR is a reproducible and inexpensive laboratory hematology index. LANR consists of three significant substances representative of the inflammatory response and nutritional factors, which include lymphocytes, neutrophils and albumin. Related studies have proven the relationship between nutrition and inflammation, finding that the systemic inflammatory response leads to a poor nutritional status and a poor prognosis for their cancer patients (*Alifano et al., 2014*). In biological terms, the imbalance in the ratio of neutrophils to lymphocytes has a potential impact on tumor progression and prognosis (*Coffelt, Wellenstein & De Visser, 2016*). Previous studies have shown that neutrophils are associated with pro-tumorigenic activities in the body, such as enhanced angiogenesis, which promotes the metastatic ability of tumor cells (*Wu et al., 2019*; *De Larco, Beverly Wuertz & Furcht, 2004*). Neutrophils are involved in different stages of the carcinogenic process, including tumorigenesis, growth, proliferation or metastatic spread (*Swierczak et al., 2015*). It promotes tumorigenesis by releasing reactive oxygen species (ROS), reactive nitrogen species (RNS) or proteases, tumor proliferation by weakening the immune system, and metastatic spread by inhibiting natural killing function and promoting tumor cell extravasation (*Xie et al., 2022*). In contradistinction, lymphocytes play an immune surveillance role in cancer, which may have anti-tumor effects and protect the host from tumor development through apoptosis-induced T-cell immune responses (*DeNardo & Coussens, 2007*). Related studies have shown that elevated neutrophils and decreased

**Table 1  Baseline clinicopathological characteristics of breast cancer patients.**

| | | Disease progression | | P* |
|---|---|---|---|---|
| | | **With** **N=69 (%)** | **Without** **N=131 (%)** | |
| Age (yr)[a] | | 59 (53-65) | 54 (49-60) | <0.001 |
| BMI | | | | 0.13 |
| | <18.5 | 2 (2.9) | 5 (3.8) | |
| | 18.5~23.9 | 32 (46.4) | 62 (47.3) | |
| | >23.9 | 35 (50.7) | 64 (48.9) | |
| Tumor size | | | | 0.03 |
| | <2 cm | 30 (43.5) | 71 (54.2) | |
| | >2 cm,<5 cm | 37 (53.6) | 59 (45) | |
| | >5 cm | 2 (2.9) | 1 (0.8) | |
| TNM stage | | | | <0.001 |
| | I | 10 (14.5) | 72 (55) | |
| | II | 23 (33.3) | 47 (35.9) | |
| | III | 36 (52.2) | 12 (9.2) | |
| Node positivity | | | | <0.001 |
| | 0 | 15 (21.7) | 111 (84.7) | |
| | 1~3 | 23 (33.3) | 15 (11.5) | |
| | 4~9 | 18 (26.1) | 3 (2.3) | |
| | >10 | 13 (18.8) | 2 (1.5) | |
| ER | | | | 0.23 |
| | Positive | 42 (60.9) | 88 (67.2) | |
| | Negative | 27 (39.1) | 43 (32.8) | |
| PR | | | | 0.54 |
| | Positive | 34 (49.3) | 64 (48.9) | |
| | Negative | 35 (50.7) | 67 (51.1) | |
| HER-2 | | | | 0.25 |
| | Positive | 54 (78.3) | 109 (83.2) | |
| | Negative | 15 (21.7) | 22 (16.8) | |
| Lym($10^9$ /L)[a] | | | | <0.001 |
| | | 1.10 (0.90-1.40) | 1.40 (1.00-1.70) | |
| Neu ($10^9$/L)[a] | | | | <0.001 |
| | | 5.30 (3.35-6.70) | 3.00 (2.20-4.90) | |
| ALB (G/L)[a] | | | | 0.001 |
| | | 38.80 (37.20-42.25) | 41.00 (39.10-43.50) | |

**Notes.**

Lym, lymphocyte; Neu, neutrophils; ALB, albumin; BMI, body mass index; ER, estrogen receptor; PR, progesterone receptor; Her-2, human epidermal growth factor receptor2; TNM, tumor-node-metastasis.

*P-values were calculated by the Student's $t$-test or Wilcoxon test for continuous variables, and the Chi-square test for categorical variables, respectively.

[a]Age, BMI, Tumor size and Node positivity, Lym, Neu , ALB, are continuous variables, the others (TNM stage, ER, PR and HER-2) are categorical variables.

lymphocytes are associated with a poorer prognosis of cancer (*Tokunaga et al., 2015*). In addition to inflammatory factors, nutritional indicators can also predict the prognosis of breast cancer, and pre-therapy serum albumin has been used as a predictor to assess disease progression, disease severity and prognosis (*Yeun & Kaysen, 1998*; *Gupta & Lis, 2010*). ALB

**Table 2  Univariate and multivariate COX regression analysis of progression-free survival in breast cancer patients.**

| Cut-point | AUC | | Univariate HR(95% CI) | P | Multivariate[a] HR(95% CI) | P |
|---|---|---|---|---|---|---|
| Lym($10^9$ /L) | 0.66 | 1.55 | 0.03 (0.586–0.737) | <0.001 | 1.756 (0.733–4.209) | 0.197 |
| Neu($10^9$ /L) | 0.70 | 4.64 | 0.038 (0.624–0.773) | <0.001 | 0.897 (0.755–1.065) | 0.240 |
| ALB(GL) | 0.65 | 38.8 | 0.043 (0.565–0.733) | 0.001 | 0.886 (0.819–0.959) | 0.002 |
| LANR | 0.75 | 12.8 | 0.035 (0.679–0.817) | <0.001 | 0.900 (0.834–0.972) | 0.007 |
| Age | | | 1.066 (1.032–1.102) | <0.001 | 1.063 (1.025–1.102) | 0.001 |
| TNM stage | | | 3.795 (2.672–5.390) | <0.001 | 5.71 (2.473–13.423) | <0.001 |
| Tumor size | | | 1.167 (1.023–1.332) | 0.022 | 0.911 (0.778–1.067) | 0.256 |
| Node positivity | | | 1.163 (1.121–1.206) | <0.001 | 1.079 (1.018–1.144) | 0.010 |

**Notes.**

HR, Hazard Ratio; CI, Confidence Interval; AUC, Area under the ROC Curve; Lym, lymphocyte; Neu, neutrophils; ALB, albumin; LANR, Lym*Alb/Neu.

[a]Multivariate cox regression models included age, TNM stage, Tumor size and Node positivity for mutual adjustment.

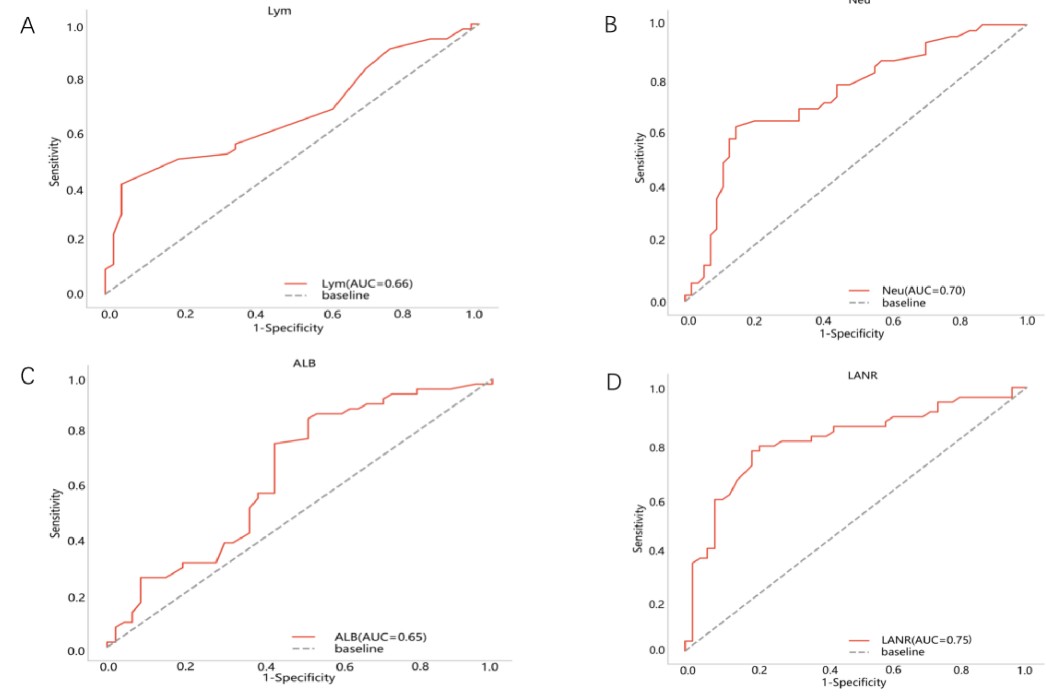

**Figure 1  Best cutoff values of preoperative LANR index, albumin, lymphocyte and neutrophils of 200 breast cancer patients.** (A) Lym for PFS. (B) Neu for PFS. (C) Alb for PFS. (D) LANR for PFS.

is mainly used for tissue repair, mainly as a carrier protein, but also to assess metabolism and immunity. Furthermore, in patients with hypoproteinemia, the development of tumor cachexia is exacerbated and the nutritional status is further worsened (*Critselis et al., 2011*). For these reasons, it is reasonable to combine the new index LANR composed of neutrophils, lymphocytes and albumin to estimate a potential indicator of individual prognosis.

**Table 3** Clinicopathological characteristics of progression-free survival in breast cancer patients in relation to LANR.

| | | LANR value | | |
| --- | --- | --- | --- | --- |
| | | **LOW** N = 87 (%) | **HIGH** N = 113 (%) | *P*-value |
| Age(mean ± SD) | | 58 (51-64) | 54 (49-61) | 0.023 |
| BMI | | | | 0.327 |
| | <18.5 | 2 (2.3) | 5 (4.4) | |
| | 18.5~23.9 | 41 (47.1) | 53 (46.9) | |
| | >23.9 | 44 (50.6) | 55 (48.7) | |
| TNM stage | | | | <0.001 |
| | I | 23 (26.4) | 59 (52.2) | |
| | II | 32 (36.8) | 38 (33.6) | |
| | III | 32 (36.8) | 16 (14.2) | |
| Tumor size | | | | 0.737 |
| | <2 cm | 47 (54.0) | 54 (47.8) | |
| | >2 cm,<5 cm | 38 (43.7) | 58 (51.3) | |
| | >5 cm | 2 (2.3) | 1 (0.9) | |
| Node positivity | | | | <0.001 |
| | 0 | 35 (40.2) | 91 (80.5) | |
| | 1~3 | 25 (28.7) | 13 (11.5) | |
| | 4~9 | 17 (19.5) | 4 (3.5) | |
| | >10 | 10 (11.5) | 5 (4.4) | |
| ER | | | | 0.232 |
| | Positive | 56 (64.4) | 74 (65.5) | |
| | Negative | 31 (35.6) | 39 (34.5) | |
| PR | | | | 0.622 |
| | Positive | 42 (48.3) | 56 (49.6) | |
| | Negative | 45 (51.7) | 57 (50.4) | |
| HER-2 | | | | 0.004 |
| | Positive | 65 (74.7) | 98 (86.7) | |
| | Negative | 22 (25.3) | 15 (13.3) | |
| PFS | | | | <0.001 |
| | Yes | 50 (57.4) | 20 (17.6) | |
| | No | 37 (42.5) | 93 (82.3) | |

**Notes.**

LANR, Lym*Alb/Neu.

*P values were calculated by the Student's *t*-test or Wilcoxon test for continuous variables, and the Chi-square test for categorical variables, respectively.

[a]Age, Tumor Size, BMI and Node positivity are continuous variable, the others (TNM stage, ER, PR and HER-2) are categorical variables.

In this study, univariate analysis showed that TNM stage, tumor size and positive axillary lymph nodes were all significantly associated with progression-free survival, whereas there was no relationship between tumor size and prognosis in the results of multifactorial analysis. In the eighth edition of the TNM classification published by the AJCC, tumor size, lymph node status, and distant metastasis were strongly associated with survival (*He et al., 2023*). The authors speculated that this result was associated with a small sample size and therefore did not enable a stratified analysis.

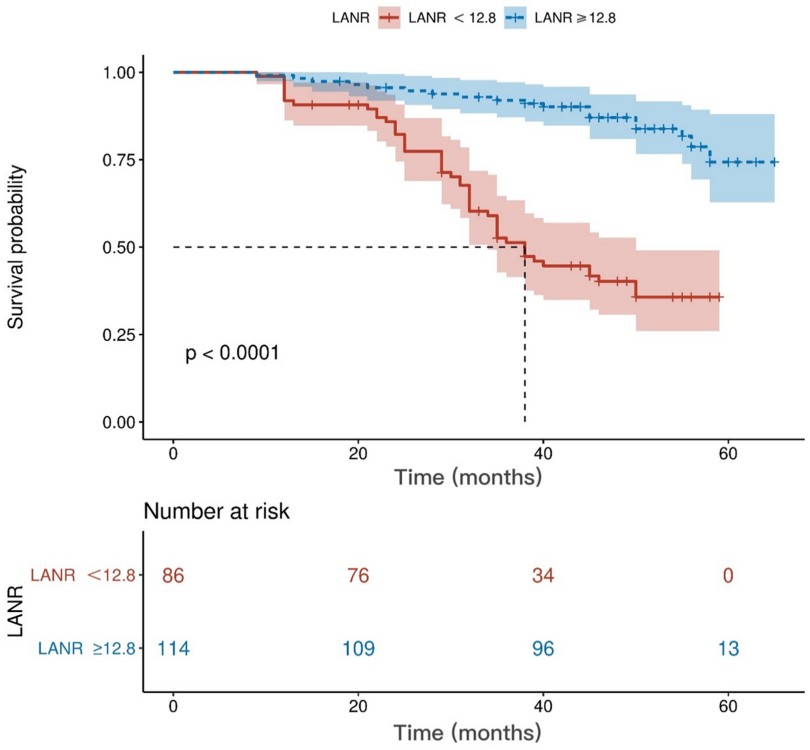

**Figure 2 Kaplan–Meier curves for progression-free survival of breast cancer patients based on LANR.**
LANR, Lym*Alb/Neu.

We investigated the relationship between clinicopathological features of breast cancer and LANR and found that age, TNM stage, positive axillary lymph nodes and human epidermal growth factor receptor 2 (c-erbB2 or HER2) were associated with LANR. According to previous studies, a 10-year increase in age was associated with a 0.8 g/L decrease in serum albumin after adjusting for health status and demographic factors (*Salive et al., 1992*). As for TNM stage, positive axillary lymph nodes and human epidermal growth factor receptor 2 (c-erbB2 or HER2), as traditional correlates of breast cancer prognosis, previous studies have found that patients with greater stage and the presence of axillary lymph node metastases may have more severe tumor malignancy, while neutrophils and lymphocytes are strongly associated with tumor progression (*Lee, Kim & Lee, 2019*); therefore, our study explains the possible reasons for the correlation of some clinicopathological features with LANR. Our ROC results demonstrated that LANR had the best area under the curve of 0.748 with an HR (95% CI) of 0.035 (0.679−0.817), and thus LANR may be a relatively good indicator of breast cancer later on. In addition, we found that patients with high levels of LANR had longer progression-free survival.

This study has several limitations. First, it was a single-center study with a small sample size (200 cases), and the findings may not be sufficient to fully reflect the status of breast cancer. Future studies should incorporate multicenter, large sample sizes to validate the role of LANR in breast cancer prognosis. Second, this is a retrospective study with certain

inherent limitations. Third, the survival endpoint was defined as progression-free survival of patients with a median follow-up time of 46 months, which is too short a follow-up time and lacks OS, and the results may differ from those studies using tumor-related death as a survival endpoint. Fourth, we evaluated LANR at only one time point before surgery. LANR changes throughout the patient's cancer treatment cycle may have a more significant impact on prognosis, but we did not evaluate it in our study.

## CONCLUSIONS

In conclusion, the data from our study confirm that a high preoperative LANR may be an easily accessible new predictor of progression-free survival in breast cancer patients. Therefore, in clinical practice, LANR may be a useful biomarker in the evaluation of breast cancer, possibly to guide appropriate treatment, adjust follow-up intervals and improve prognosis.

### Funding
This work was supported by the Top Talent Support Program for Young and Middle-aged people of Wuxi Health Committee (BJ2020047), the Mading the Taihu Lake Talent Plan (4532001THMD), and the Beijing Bethune charitable Foundation (2022-YJ-085-J-Z-ZZ-011). The funders had no role in study design, data collection and analysis, decision to publish, or preparation of the manuscript.

### Grant Disclosures
The following grant information was disclosed by the authors:
Top Talent Support Program for Young and Middle-aged people of Wuxi Health Committee: BJ2020047.
Taihu Lake Talent Plan: 4532001THMD.
Beijing Bethune charitable Foundation: 2022-YJ-085-J-Z-ZZ-011.

### Competing Interests
The authors declare there are no competing interests.

### Author Contributions
- Yuan Wang conceived and designed the experiments, performed the experiments, analyzed the data, prepared figures and/or tables, and approved the final draft.
- Jiaru Zhuang performed the experiments, analyzed the data, prepared figures and/or tables, and approved the final draft.
- Shan Wang performed the experiments, analyzed the data, prepared figures and/or tables, and approved the final draft.
- Yibo Wu conceived and designed the experiments, authored or reviewed drafts of the article, and approved the final draft.
- Ling Chen conceived and designed the experiments, authored or reviewed drafts of the article, and approved the final draft.

## Human Ethics

The following information was supplied relating to ethical approvals (*i.e.*, approving body and any reference numbers):

This study was approved by the Medical Ethics Committee of the Affiliated Hospital of Jiangnan University (JNMS04202301072)

## Data Availability

The raw data is available in the Supplemental File.

## Supplemental Information

Supplemental information for this article can be found online at http://dx.doi.org/10.7717/peerj.17382#supplemental-information.

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
