# Peer review of "The prognostic value of preoperative neoindices consisting of lymphocytes, neutrophils and albumin (LANR) in operable breast cancer: a retrospective study"

_PeerJ, doi:10.7717/peerj.17382_

## Round 0.1 · original submission · Major Revisions

Dear authors, thank you for your submission. Your work requires a few revisions that are considered substantial. Please, refer to the reviewers' comments for further details.

Reviewer 1 ·

Basic reporting

This study sheds light on the prognostic significance of LANR in breast cancer patients, offering a well-organized analysis. However, making the statistical methods clearer, enhancing the readability of visuals, refining the logic behind the research, and cautiously affirming the conclusions would greatly improve the paper.

Experimental design

1. The paper effectively employs ROC curves, Cox proportional risk regression, and subgroup analysis. Yet, it doesn't fully explain why the inflection point was chosen to switch continuous variables to dichotomous. A deeper explanation, with references to similar methods in past studies, would solidify the study's approach.
2. The explanation of the hazard ratio (HR) for LANR, particularly its protective effect suggested by an HR of 0.035 (0.679-0.817), needs more clarity. Discussing how LANR values relate to patient outcomes in more detail would enrich its prognostic value.
3. Kaplan-Meier and ROC curves play a crucial role in presenting the findings, but the legends and labels need to be clearer. Enlarging font sizes and using contrasting colors for better distinction is advisable.

Validity of the findings

1. The discussion strongly links LANR with its prognostic importance. However, a comparison with or consideration of integration into current prognostic models could provide a broader perspective on LANR's role in breast cancer prognosis.
2. The conclusion summarizes the results well, but it could echo a more cautious tone about the limits of retrospective studies and the necessity for future validation. Recognizing these points would enhance the conclusion's impact and reliability.

Reviewer 2 ·

Basic reporting

• The article has been written in clear, unambiguous and profession English Language throughout.
• Additional references will need to be added in the Introduction to provide proper background and literature supporting the context.
• The structure of the article complies with PeerJ standards.
• Figures are relevant, well-labelled but needs better description.
• Raw data has been supplied.
• Write your in-text citation before full stop. For instance, in line 44 - 45: Breast cancer is an issue of public health significance and is the most commonly diagnosed malignancy in women throughout the world (Ferlay et al. 2015).
• The citations in the References section should appear in accordance with the order in which the information from that document appears in the main body. For instance: In Introduction section, the information from Ferlay et al. 2015, Sung et al. 2021, Shankaran et al. 2001 were cited in 1st, 2nd and 3rd order, respectively. In References, their reference should be in the same order as well (1st, 2nd and 3rd ).
• Line 50-53 of Introduction does not seem coherent and complete. They need further modification.
• At the beginning of line 51, there are extra spaces.
• Line 55-56 need proper referencing that relates nutrition, inflammation and cancer prognosis.
• Additional facts and citations are suggested to add in the second paragraph of Introduction. This article may help in this: Nomograms based on the lymphocyte–albumin–neutrophil ratio (LANR) for predicting the prognosis of nasopharyngeal carcinoma patients after definitive radiotherapy.
• Briefly mention the method and outcome of the study in Introduction following line 73.
• Write the full form of PFS in line 100 of Data Collection.
• Incorrect line spacing in Conclusion and Acknowledgement.
• Figure 1 legend needs to be more descriptive. Label the 4 graphs as a, b, c, d. Describe which graph refers to what in figure legend.
• Need more description in figure legend of Figure 2.

Experimental design

• This is original primary research within the scopes of the journal.
• Research question is well defined, relevant & meaningful. It is stated how the research fills an identified knowledge gap.
• Rigorous investigation has been performed to a high technical & ethical standard.
• Methods described with sufficient detail & information to replicate.

Validity of the findings

• All underlying data have been provided; they are robust, statistically sound, & controlled.
• Conclusions are well stated, linked to original research question & limited to supporting results.

---

## Round 0.2 · Minor Revisions

Dear authors, many thanks for your resubmission. Please, proceed with minor revisions as per reviewers' comments. Particularly, constructing proper figures' and tables' legends. This is an important work, and commendable improvements have been done. Make sure to proofread the whole document before resumission, meeting the journals requirements particularly in figures's quality. Looking forward for your revised version.

Reviewer 1 ·

Basic reporting

none

Experimental design

NONE

Validity of the findings

none

Reviewer 2 ·

Basic reporting

• The article has been written in clear, unambiguous and profession English Language throughout.
• Additional references will need to be added in the Introduction to provide proper background and literature supporting the context.
• The structure of the article complies with PeerJ standards.
• Figures are relevant, well-labelled but needs better description.
• Raw data has been supplied.
• Line 64-67 needs to be paraphrased.
• The citations in the References section should appear in accordance with the order in which the information from that document appears in the main body. For instance: In Introduction section, the information from Ferlay et al. 2015, Sung et al. 2021, Shankaran et al. 2001 were cited in 1st, 2nd and 3rd order, respectively. In References, their reference should be in the same order as well (1st, 2nd and 3rd ). Use endnote to make this easier.
• There should be a Full stop in line 147.
• Figure legends need to be more descriptive.

Experimental design

• This is original primary research within the scopes of the journal.
• Research question is well defined, relevant & meaningful. It is stated how the research fills an identified knowledge gap.
• Rigorous investigation has been performed to a high technical & ethical standard.
• Methods described with sufficient detail & information to replicate.

Validity of the findings

• All underlying data have been provided; they are robust, statistically sound, & controlled.
• Conclusions are well stated, linked to original research question & limited to supporting results.

---

## Round 0.3 · accepted · Accept

I am happy to let you know that i am now accepting your manuscript for publication in PeerJ. Many thanks for your hardwork and submission.